# Some Interesting Observations on the Free Energy Principle

**DOI:** 10.3390/e23081076

**Published:** 2021-08-19

**Authors:** Karl J. Friston, Lancelot Da Costa, Thomas Parr

**Affiliations:** 1The Wellcome Centre for Human Neuroimaging, University College London, London WC1N 3AR, UK; k.friston@ucl.ac.uk (K.J.F.); l.da-costa@imperial.ac.uk (L.D.C.); 2Department of Mathematics, Imperial College London, London SW7 2AZ, UK

**Keywords:** free energy principle, variational, Bayesian, Markov blanket

## Abstract

Biehl et al. (2021) present some interesting observations on an early formulation of the free energy principle. We use these observations to scaffold a discussion of the technical arguments that underwrite the free energy principle. This discussion focuses on solenoidal coupling between various (subsets of) states in sparsely coupled systems that possess a Markov blanket—and the distinction between exact and approximate Bayesian inference, implied by the ensuing Bayesian mechanics.

## 1. Introduction

We enjoyed reading the deconstruction of the free energy principle (FEP) in [1]—as introduced some years ago in [2]. Having said this, no one likes to be told that they have made mistakes. Fortunately, all of the observations in [1] are interesting, some are correct and none confound the FEP. In what follows, we use the observations of Biehl et al. (ibid) to drill down on the interesting points they raise—and their implications in the setting of the FEP.

To contextualise these observations, we first rehearse the major steps in deriving the FEP and then focus on three cardinal issues addressed in Biehl et al.; namely, what is the precise form of the dynamical coupling among (subsets of) states that constitute a Markov blanket partition? What implications attend a nonzero evidence bound, when interpreting self-organisation as self-evidencing (i.e., Bayesian inference)? Further, when do variational free energy gradients vanish? The first of the three issues appears in Biehl et al. distributed across their observations 1–3. The second and third appear in observation 5 and the surrounding discussion. Biehl et al. make several observations; however, some are recapitulated (e.g., in the context of generalised coordinates of motion). Their observation 6 is one example of this. We ignore these observations. Please note that the numbering of observations from Biehl et al. refers to the numbers that are assigned in the main text of the paper, and not to the order in the bullet-pointed list provided at the start of the paper.

One could read Biehl et al. as a critique of early formulations of the FEP—in terms of implicit assumptions and incomplete (heuristic) proofs—as opposed to a critique of the FEP per se. However, the issues they identify are still fundamental. Some of these issues are addressed in [3]. However, that monograph has not been subject to external peer review (and contains at least one technical error). A concise version of the Bayesian mechanics is presented in [4]. In what follows, we will use the notation and nomenclature in [3], which is currently the most comprehensive treatment of the FEP and to which we refer readers for detailed applications to physical systems. The novel contribution of this paper is an explicit specification of the conditions imposed upon a dynamical flow that are sufficient to ensure a Markov blanket.

## 2. The Free Energy Principle in Brief

Technically, the free energy principle asserts that any “thing” that attains a nonequilibrium steady state can be construed as performing an elemental sort of Bayesian inference. Informally, this can be described as self-evidencing [5]; namely, anything that exists is actively seeking evidence for its existence. In short, life is its own existence proof [6]. This self-evidencing gloss is licensed by the fact that the gradient flows that underwrite nonequilibrium steady-state can be expressed as a gradient descent on surprisal or self-information [2]. This is mathematically equivalent to a gradient ascent on log model evidence or marginal likelihood. Unpacking this further, if we take the view that (persistent) existence implies a steady-state density, interpretable as the marginal likelihood of an implicit model, the dynamical flows that underwrite this steady-state are towards regions of high probability density. In Bayesian statistics, a marginal likelihood is known as model evidence. The implication is that dissipative flows that maximise the evidence for a model—i.e., the steady-state density that operationalises existence—can be read as “seeking evidence” for existence. This provides the formal underpinning for a teleological framing of “self-organisation” in dissipative structures as “self-evidencing”. Technically, the argument involves two moves:

First, a thing is defined stipulatively in terms of a Markov blanket [7,8], such that something’s internal states are independent of its external states, when conditioned on its blanket states. Blanket states can be further partitioned into active and sensory states that are not influenced by internal and external states, respectively. This partition is not part of the definition of a Markov blanket but describes a way of characterising the blanket states. For any given blanket, the set of active or sensory states may be empty. As we will see, the dynamical coupling between these blanket states and the internal and external states is asymmetrical. However, the asymmetry emerges specifically under the assumption of sparse coupling. By starting from a Langevin formulation of a random dynamical system, the associated density dynamics can be expressed as the solutions to the Fokker Planck equation [9,10]. Crucially, because we are (stipulatively) assuming nonequilibrium steady-state, the steady-state solution to the Fokker Planck system enables us to express the dynamics or flow of states in terms of a Helmholtz decomposition [11,12,13]. This decomposition divides flow into dissipative gradient flows on the self-information of any state (i.e., its negative log probability at steady state) and a divergence-free or solenoidal flow. When this solution may be decomposed according to a Markov blanket partition, the implicit conditional dependencies require certain solenoidal coupling terms to disappear. This means that one can express the dynamics of something’s autonomous states (i.e., internal and active states) as a function of—and only of—the blanket and internal states. By construction, this is a (generalised) gradient flow on self-information or surprisal. See Figure 1.

The second move is to note that the conditional independencies, implied by the Markov blanket, induces a particular kind of information geometry [14,15] in the internal state space. In brief, for any blanket state, there must be an expected internal state—and a conditional density over external states, given that blanket state. This means there is a statistical manifold in the internal state-space, corresponding to the conditional expectations of internal states, given blanket states. In other words, every point on the internal manifold corresponds to a conditional density—or posterior Bayesian belief—over external states. This endows the internal manifold with an information geometry, where the distance between probability distributions can be measured with the Fisher information metric tensor [16,17,18], supplied by the conditional distribution over external states. Put simply, flows on the internal manifold can be construed as belief updating or Bayesian inference [19]. This view is licensed by the fact that the (average) flow on the internal statistical manifold is a gradient flow on surprisal (i.e., the negative Bayesian model evidence) of the blanket (and internal) states. In essence, this is the free energy Lemma [3].

One can take this further and use arguments related to integral fluctuation theorems [20,21], a set of results based upon characterising the probabilities of alternative paths a system might follow, to derive probability densities over the trajectory of active states. This allows one to characterise autonomous behaviour (i.e., the dynamics of autonomous states) in terms of constructs from psychology and economics; e.g., risk in relation to the goal states encoded by the steady-state density [22,23]. This affords a description of self-organisation to nonequilibrium steady state as Bayesian inference (a.k.a., active inference) that has both sentient (i.e., inferring external states of affairs) and enactive (i.e., densities over trajectories of active states) aspects. In short, the FEP furnishes a description of the perception–action cycle [24] as evinced in anything that exists, in virtue of possessing a Markov blanket.

In and of itself, this is just a theoretical exercise. Practically, things get more interesting when we use the free energy principle to engineer gradient flows by writing down a generative model, under which model evidence can be evaluated. This means one can then solve the equations of motion—that emulate the gradient flows above—to create systems that self-organise to some nonequilibrium steady-state. In this setting, the steady state is operationally defined in terms of the priors over external or blanket states that are part of the generative model. Having said this, the application of the free energy principle and active inference [25,26,27,28] is beyond the scope of the critique in Biehl et al. (ibid). They focus on the tenets that underwrite the free energy Lemma. We now turn to three key tenets, highlighted by [1].

## 3. Observation One


*Certain solenoidal coupling terms are precluded when a Markov blanket emerges under sparse coupling.*


This observation speaks to the constitution of the flow of systemic states x=(η,s,a,μ) at nonequilibrium steady state. We refer to the constituents of *x* as external, sensory, active, and internal states, respectively. This flow can be expressed as the solution to the Fokker Planck equation, in terms of a Helmholtz decomposition–also known as the fundamental theorem of vector calculus.:(1)$$\begin{array}{c} \dot{x}=f\left(x\right)+\omega \\ \;\;\;\;f\left(x\right)=\left(Q-\Gamma \right)\nabla ℑ\left(x\right) \end{array}$$

This decomposition is at the heart of the free energy principle and most formulations of nonequilibrium steady state in nonlinear systems; ranging from molecular interactions through to evolution: see [10,11,29,30]. For a concise derivation of Equation (1), under simplifying assumptions, please see Lemma D.1 in [13]. Here, ℑ(x)=−lnp(x) is surprisal or self-information and the antisymmetric matrix $Q=-Q^{T}$ mediates solenoidal flow. The density *p*(*x*) is the steady state density. The positive definite matrix Γ∝I (the identity matrix) plays the role of a diffusion tensor or covariance matrix describing the amplitude of random fluctuations, ω (assumed to be a Wiener process). In this form, the flow can be decomposed into dissipative gradient flows −Γ∇ℑ and divergence-free or solenoidal flow Q∇ℑ. Note that the off-diagonal terms of Γ are zero because random fluctuations are independent. However, this independence does not preclude solenoidal coupling among states. For simplicity, in (1) and what follows, we assume that the amplitude of fluctuations and solenoidal flow matrices vary sufficiently slowly with *x* that they can be considered constant.

Some readers may have encountered the Helmholtz decomposition in 3-dimensional spaces, where it is expressed in terms of the curl of a vector potential and the gradient of a scalar potential [31]. This form emphasises the relationship with Maxwell’s equations in electromagnetics [32], which decompose vector fields into their electric and magnetic components. While Equation (1) is expressed only in terms of a scalar potential, it is simple to rewrite this (for a 3-dimensional system) to include a vector potential (*A*):(2)$$\begin{array}{cc} f\left(x\right) & =\nabla \times A\left(x\right)-\Gamma \nabla ℑ\left(x\right)\\ A\left(x\right) & ≜\left[\begin{array}{c} -Q_{32}\\ Q_{31}\\ -Q_{21} \end{array}\right]ℑ\left(x\right)\\ Q & =\left[\begin{array}{ccc} 0 & -Q_{21} & -Q_{31}\\ Q_{21} & 0 & -Q_{32}\\ Q_{31} & Q_{32} & 0 \end{array}\right] \end{array}$$

This exploits the skew-symmetry of the *Q* matrix. The question now is which solenoidal coupling terms are consistent with the conditional independencies implied by a Markov blanket? More generally, we want to know the form of the steady state flow that engenders conditional independencies between external and internal states. Technically, a Markov blanket is the set of variables that, if known, render two other sets conditionally independent [33]. This implies the joint density of internal and external states, conditioned upon blanket states, factorizes as follows:(3)(μ⊥η)|b⇔p(μ,η|b)=p(μ|b)p(η|b)

The identification of Markov blankets at nonequilibrium steady state is not as straightforward as it might appear (please see Appendix A and Appendix B for a detailed account). Given a joint density over a partition of states, a Markov blanket comprises the parents, the children and the parents of the children of any random variable. In dynamical systems, the joint density is a function of time. This means that, to evaluate a probability density, we need to do more than specify the internal, external, and blanket states at which the density is evaluated. We also need to specify the time at which it is evaluated. For Markovian systems, the states at the current time are the blanket states that separate states in the future from states in the past. However, these are not Markov blankets of the steady-state density. This joint density rests on the solution to the density dynamics in Equation (1). Note that the usual rules of identifying parents, children and parents of children cannot be applied directly to the dynamical coupling or flow. Differentiating this solution, with respect to the states, reveals the relationship between the flow—specified by a Jacobian J=∇f(x)—and conditional independencies—specified by a Hessian $H=\nabla^{2}ℑ$:(4)$$\begin{array}{l} \;\nabla f(x)=(Q-\Gamma )\nabla^{2}ℑ(x)\Rightarrow \\ \;J(x)=(Q-\Gamma )H(x)\\ \left[\begin{array}{cccc} J_{\eta \eta } & J_{\eta s} & J_{\eta a} & J_{\eta \mu }\\ J_{s\eta } & J_{ss} & J_{sa} & J_{s\mu }\\ J_{a\eta } & J_{as} & J_{aa} & J_{a\mu }\\ J_{\mu \eta } & J_{\mu s} & J_{\mu a} & J_{\mu \mu } \end{array}]=[\begin{array}{cccc} Q_{\eta \eta }-\Gamma_{\eta \eta } & Q_{\eta s} & Q_{\eta a} & Q_{\eta \mu }\\ -Q_{\eta s}^{T} & Q_{ss}-\Gamma_{ss} & Q_{sa} & Q_{s\mu }\\ -Q_{\eta a}^{T} & -Q_{sa}^{T} & Q_{aa}-\Gamma_{aa} & Q_{a\mu }\\ -Q_{\eta \mu }^{T} & -Q_{s\mu }^{T} & -Q_{a\mu }^{T} & Q_{\mu \mu }-\Gamma_{\mu \mu } \end{array}][\begin{array}{cccc} H_{\eta \eta } & H_{\eta s} & H_{\eta a} & H_{\eta \mu }\\ H_{\eta s}^{T} & H_{ss} & H_{sa} & H_{s\mu }\\ H_{\eta a}^{T} & H_{sa}^{T} & H_{aa} & H_{a\mu }\\ H_{\eta \mu }^{T} & H_{s\mu }^{T} & H_{a\mu }^{T} & H_{\mu \mu } \end{array}\right] \end{array}$$

Here, the flow constraints are summarized to the first order by the Jacobian. For example, if the Jacobian encoding the coupling between external and internal states is zero, we can express the flow of internal states as a function of, and only of, particular states (*π*). Particular states comprise internal states and their Markov blanket. These can be construed as the states of a particle; hence, particular states:(5)$$J_{\mu \eta }=\nabla_{\eta }f_{\mu }\left(x\right)=0\Rightarrow f_{\mu }\left(x\right)=f_{\mu }\left(\pi \right)$$

Similarly, the Hessian or curvature matrix encodes conditional dependencies, in the sense that if the corresponding submatrix is zero, internal and external states are conditionally independent:(6)$$H_{\mu \eta }=\nabla_{\mu \eta }ℑ\left(x\right)=0\Rightarrow ℑ(\mu |b,\eta )=ℑ(\mu |b)\Rightarrow \left(\mu ⊥\eta \right)|b$$

Equation (4) shows that the amplitude of random fluctuations and solenoidal coupling play a key role in relating flow constraints and conditional dependencies. The solenoidal components are especially important in the setting of nonequilibrium steady state. Indeed, on one reading of nonequilibrium dynamics, the very presence of solenoidal flow is sufficient to break detailed balance—and preclude any equilibria in the conventional (statistical mechanics) sense [10,21,31,34].

The question now is which solenoidal terms and conditional independencies admit a Markov blanket. We are interested in Markov blankets that emerge from sparse coupling among states. Here, sparse coupling is taken to mean that no state is influenced by all other states. Clearly, the flows can be fine-tuned to create Markov blankets in the absence of any sparsity constraints on coupling (for examples of this fine-tuning, see Appendix A of Biehl et al. [1]); however, the FEP only applies to Markov blankets that emerge under sparse flows; in particular, when autonomous states are uncoupled from external states (by definition). This was not made explicit in early formulations of the free-energy principle, which focused on simple cases that satisfied this condition. Much of the critique in Biehl et al. is completely understandable in light of this omission. Condition 1 in Biehl et al. satisfies the flow constraint, while Condition 2 corresponds to the existence of a Markov blanket. Biehl et al. then observe that Condition 1 does not imply Condition 2, and vice versa (Observation 1). This is generally true; however, the FEP only applies when Condition 2 is satisfied under Condition 1. The issue at hand is to identify the functional forms of steady-state flow that satisfy both conditions. In other words, each row of the Jacobian must contain at least one zero entry. Inspection of (4) shows that this can only be satisfied when each row of the Hessian contains at least one zero entry. If we supplement the requisite conditional independence between internal and external states with conditional independencies between active and external states—and between sensory and internal states—we have the following functional form (please see Appendix A and Appendix B for a more formal analysis):(7)$$\left[\begin{array}{cccc} J_{\eta \eta } & J_{\eta s} & J_{\eta a} & \\ J_{s\eta } & J_{ss} & J_{sa} & \\ & J_{as} & J_{aa} & J_{a\mu }\\ & J_{\mu s} & J_{\mu a} & J_{\mu \mu } \end{array}\right]=\left[\begin{array}{cccc} Q_{\eta \eta }-\Gamma_{\eta \eta } & Q_{\eta s} & & \\ -Q_{\eta s}^{T} & Q_{ss}-\Gamma_{ss} & & \\ & & Q_{aa}-\Gamma_{aa} & Q_{a\mu }\\ & & -Q_{a\mu }^{T} & Q_{\mu \mu }-\Gamma_{\mu \mu } \end{array}\right]\left[\begin{array}{cccc} H_{\eta \eta } & H_{\eta s} & & \\ H_{\eta s}^{T} & H_{ss} & H_{sa} & \\ & H_{sa}^{T} & H_{aa} & H_{a\mu }\\ & & H_{a\mu }^{T} & H_{\mu \mu } \end{array}\right]$$

There are other conditional independence structures that one could consider. Interesting examples include circular flow constraints (when external states influence sensory states that influence internal states that influence active states that influence external states) or flow constraints that lead to conditional independence between internal and external states—and between sensory and active states. This latter case was used by Biehl et al. (Appendix A and Appendix B), in their so-called counterexamples; however, these are simply counterexamples to the functional form of Equation (7), not counterexamples that violate the assumptions of the free energy lemma. The form in Equation (7) satisfies the sparse coupling constraint by precluding solenoidal coupling between autonomous and non-autonomous states:(8)$$\left[\begin{array}{cc} Q_{\eta a} & Q_{\eta \mu }\\ Q_{sa} & Q_{s\mu } \end{array}\right]=0\Rightarrow \left\{\begin{array}{c} J_{\eta \mu }=Q_{\eta a}H_{a\mu }+Q_{\eta \mu }H_{\mu \mu }\\ J_{s\mu }=Q_{sa}H_{a\mu }+Q_{s\mu }H_{\mu \mu }\\ J_{a\eta }=-Q_{sa}^{T}H_{\eta s}^{T}-Q_{\eta a}^{T}H_{\eta \eta }\\ J_{\mu \eta }=-Q_{s\mu }^{T}H_{\eta s}^{T}-Q_{\eta \mu }^{T}H_{\eta \eta } \end{array}\right\}=0$$

There are three special cases of (7) that obtain when suppressing (i.e., setting to zero) solenoidal coupling between sensory and external states, between active and internal states or both. The latter case was considered in [2], in which there is no solenoidal coupling between the different kinds of states. This special case is referred to as Condition 3 in Biehl et al. (ibid). This special case may be ubiquitous in systems with short-range coupling; for example, the simple cell-like structure in the lower panel of Figure 1. In this simple case, active states are effectively shielded from external states by sensory states, while sensory states are separated from internal states by active states. The general case corresponds to the upper panel in Figure 1, in which active states influence external states directly—and sensory states are coupled directly to internal states. In both instances, autonomous states are functions of, and only of, particular states and internal states are conditionally independent of external states. These two conditions underwrite the free energy lemma below.

In summary, substituting (8) into (1) allows us to express the flow of autonomous states as a function of, and only of, particular states:(9)$$\left[\begin{array}{c} f_{\eta }\left(\eta ,b\right)\\ f_{s}\left(\eta ,b\right)\\ f_{\alpha }\left(\mu ,b\right) \end{array}\right]=\left[\begin{array}{ccc} Q_{\eta \eta }-\Gamma_{\eta \eta } & Q_{\eta s} & \\ -Q_{\eta s}^{T} & Q_{ss}-\Gamma_{ss} & \\ & & Q_{\alpha \alpha }-\Gamma_{\alpha \alpha } \end{array}\right]\left[\begin{array}{c} \nabla_{\eta }ℑ\\ \nabla_{s}ℑ\\ \nabla_{\alpha }ℑ \end{array}\right]\Rightarrow \left(\mu ⊥\eta \right)|b$$

Recall that *x* is a vector quantity, so *f* is a vector-valued function. The subscripted indices in (9) identify elements of this vector. Please refer to Figure 1 for the sets of states indicated by each subscripted symbol. This (sparse) flow precludes solenoidal coupling between autonomous and non-autonomous states and renders internal and external states conditionally independent. However, it does not preclude solenoidal coupling between internal and active states—or between external and sensory states. Examples of these forms of solenoidal coupling may be commonplace. For example, the solenoidal coupling between external and sensory states may be manifest in oscillatory interactions between the external milieu and sense organs that respond to vibrations (e.g., the tympanic membrane of the ear). Similarly, solenoidal coupling between internal and active states may be ubiquitous in sentient creatures with brains—in the form of pacemakers and central pattern generators that produce stereotyped behaviors, such as phonation or respiration [35].

More generally, solenoidal coupling may be essential for self-organisation and active inference; especially, when considered in the light of oscillations and communication [34] or, indeed, evolutionary dynamics: at an evolutionary level, solenoidal flows and fluxes play a central role in accounting for self-organisation in evolution in terms of phenomena like Red Queen dynamics [10]. Red Queen dynamics can be interpreted in two (related) ways that relate to the solenoidal and dissipative parts of the Helmholtz decomposition. The first is that, in systems with small amplitude fluctuations, the solenoidal flow causes the system to revisit the same regions repeatedly. The second is that the presence of large-amplitude random fluctuations would cause diffusion of the probability density if it were not for the dissipative part of the flow. In other words, at steady state, the dissipative flow counteracts the random fluctuations to keep the density from changing; namely, the endless co-evolution that persists at evolutionary steady-state, following the optimization of fitness; i.e., surprisal or, in the treatment of [30], intrinsic potential.

## 4. Observation Two


*The difference between the variational density and conditional density, as assessed by a Kullback-Leibler (KL) divergence or bound can be arbitrarily large.*
$$D[q_{\mu }\left(\eta \right)||p(\eta |b)]=c\geq 0$$


This divergence was characterized as (less than) *c* in [1] (there listed as observation 5 (iii)), who note that *c* does not have an upper bound. So, what does this imply for the free energy lemma? In itself, this observation is unremarkable. In variational inference, the KL divergence is used to construct an upper bound to surprise, implying it is only necessary for the divergence to have a lower bound. However, it brings into focus a key distinction between different interpretations of the Bayesian mechanics implied by the information geometry endowed by a Markov blanket. Recall from above that there exists a statistical manifold in the internal state space, on which the flow of (conditional expectations of) internal states $\mu \left(b\right)=E_{p}[\mu |b]$ perform a gradient descent on the surprisal of particular states. Note that we are dealing with conditional expectations of internal states, denoted by boldface. The derivations in Biehl et al. and subsequent observations (e.g., observations 4) ignore this definition of the internal manifold—and can therefore be discounted. They overlooked this because their arguments are based on the heuristics in [2]. Similarly, ref. [36] failed to specify this important aspect of the free energy lemma. Please see the Appendix C. The gradient descent on surprisal can be expressed as a gradient flow on a free energy function of particular states π=(μ,b). From Equation (9)
(10)$$\begin{array}{c} f_{\alpha }\left(\mu ,b\right)=\left(Q_{\alpha \alpha }-\Gamma_{\alpha \alpha }\right)\nabla_{\alpha }ℑ\left(\mu ,b\right)\\ \;\;\;\;\;\;\;\;\;\;\;\;\;\;\;\;\approx \left(Q_{\alpha \alpha }-\Gamma_{\alpha \alpha }\right)\nabla_{\alpha }F\left(\mu ,b\right) \end{array}$$

The approximation in (10) holds when the divergence varies sufficiently slowly with changes in particular states. This means the most likely path conforms to a variational principle of least action, where the free energy $F\left(\pi \right)\equiv F\left[q_{\mu }\left(\eta \right),\pi \right]$ is a functional of a variational density $q_{\mu }\left(\eta \right)\approx p(\eta |b)$ that is parameterized by the conditional expectation of internal states. In general, this free energy is an upper bound on the surprisal of particular states:(11)$$\begin{array}{ll} F(\pi ) & ≜\underset{\mathrm{energy}}{\underset{︸}{E_{q}[ℑ(\eta ,\pi )]}}-\underset{\mathrm{entropy}}{\underset{︸}{H\left[q_{\mu }(\eta )\right]}}\\ & =\underset{\mathrm{surprisal}}{\underset{︸}{ℑ(\pi )}}+\underset{\mathrm{evidence}\;\mathrm{bound}}{\underset{︸}{D\left[q_{\mu }(\eta )∥p(\eta |b)\right]}}\\ & =\underset{\mathrm{inacuracy}}{\underset{︸}{E_{q}[ℑ(\pi |\eta )]}}+\underset{\mathrm{complexity}}{\underset{︸}{D\left[q_{\mu }(\eta )∥p(\eta )\right]}}\geq ℑ(\pi ) \end{array}$$

The expectations in (11), including those implicit in the KL divergences, relate to the external states only. This is evident in that the free energy is a function of particular states and not their average. This functional can be expressed in several forms; namely, expected energy minus the entropy of the variational density, which is equivalent to the self-information associated with particular states (i.e., *surprisal*) plus the KL divergence between the variational and posterior density (i.e., *evidence bound*). In turn, this can be decomposed into the expected negative log-likelihood of particular states (i.e., [*in*]*accuracy*) and the KL divergence between posterior and prior densities (i.e., *complexity*). In short, free energy constitutes a *Lyapunov function* for the expected flow of autonomous states.

This functional is referred to as a free energy because it comprises an entropy and an expected potential [37]. However, it can be re-expressed as the surprisal associated with particular states (itself interpretable as a potential) and the KL divergence (i.e., relative entropy) between the variational density—parameterized by any point on the internal manifold—and the posterior over external states, given blanket states. In this instance, the expected potential is the surprisal and the KL divergence is between the variational density—parameterized by any point on the internal manifold—and the posterior over external states, given blanket states. In Bayesian statistics and machine learning, this divergence is known as an evidence bound. This is because it supplies a non-negative bound on surprisal or log evidence [19,38,39]. To license an interpretation of surprisal, in terms of model evidence, we have to express surprisal in terms of a generative model; namely, a joint density over causes and consequences. Here, the generative model is simply the steady-state density over external (causes) and blanket (consequences) states.

In summary, the existence of a Markovian partition allows one to express dynamics as either gradient flows on (i) the NESS potential or (ii) variational free energy. These alternative formulations mean that one can interpret (physical) dynamics as (inferential) processes that minimise variational free energy. Technically, the NESS potential (i.e., surprisal) is a function of particular states. Conversely, variational free energy can be read as a functional of a probability density over external states, which is parameterised by internal states. This equips the gradient flows with an information geometry that licences a teleological interpretation of dynamics in terms of inference. Put simply, one can either regard a “thermometer” as (i) a physical system responding to forces and thermodynamic fluxes or, (ii) measuring (i.e., inferring) the ambient (i.e., external) temperature.

Focusing on this example, one could formulate the behaviour of a (mercury) thermometer in terms of a partition of states. These include the ambient temperature (external), the temperature of the mercury in the bulb (sensory), the volume of the mercury (active), and the height of the column of mercury in the cylinder (internal). Note that this system satisfies the conditions outlined in observation 1 as, if we knew the volume of mercury, the ambient temperature would tell us nothing new about the height of the column—satisfying the Markov blanket condition. Furthermore, the sparse coupling condition is satisfied as there is no direct dynamical coupling between the internal and external states. To define a variational density in this setting, we would note that, given the temperature and volume of the mercury, there is a well-defined mapping between the expected height of the mercury and the distribution of ambient temperatures we would expect. Expressing this distribution as a function of the conditional expectation of the height of the mercury gives us the variational density.

The thermometer example illustrates the importance of the result in (10). If we wanted to build a thermometer, we would not start from the equations of motion and accompanying NESS density, we would start with a generative model and perform a gradient descent on the accompanying variational free energy. In other words, the importance of the FEP lies in formulating (active) inference as a principle of least action that can be realised in physical systems or in silico, given a generative model.

### Exact or Approximate?

There are two ways that we can take this interpretation of flows on the internal (statistical) manifold forward. The first and simplest is to stipulate that the variational density—parameterized by the internal state or coordinate on an internal manifold—is the posterior over external states. On this view, the bound in (11) collapses to zero and the flow of (the conditional expectations, given blanket states, of) internal states can be read as performing exact Bayesian inference. However, this interpretation fails to specify how the (conditional expectations of) internal states parameterize the posterior Bayesian beliefs over external states. To do this, we would need to define a functional form for the variational density and associate internal states with its parameters or sufficient statistics (e.g., mean and precision). However, as soon as we commit to a parameterized form, we move away from *exact Bayesian inference* and into the realm of *approximate Bayesian inference*. This is because the exact equivalence between the variational and posterior density over external states is no longer guaranteed. This inflates the bound above, leading to variational Bayes.

This kind of inference predominates in the statistical and machine learning literature, because it is relatively straightforward to compute the variational free energy, given a parameterized form for the variational density [38,39,40]. On the other hand, it is practically impossible to evaluate the Bayesian model evidence directly. On some accounts of variational Bayes, the use of a variational free energy was introduced by Richard Feynman in the setting of the path integral formulation of quantum electrodynamics [37]. Effectively, it converts an intractable integration problem into a tractable optimization problem. In other words, it affords a computable objective function, whose minimization will approximate the minimization of the evidence, which is always upper bounded by the variational free energy. Indeed, in machine learning, variational free energy is often referred to as an evidence lower bound (ELBO) [19].

From our perspective, this means that we can either interpret the flow on an internal manifold as exacting exact Bayesian inference. Alternatively, if we committed to a functional form for the variational density, it would look as if the flow is approximating approximate Bayesian inference. This distinction was articulated in terms of the difference between a *particular* and *variational* free energy in [3]. The functional form for the variational density was Gaussian, leading to a ubiquitous form of variational Bayes under the Laplace assumption [41]. The details here are not important. The key thing to observe is that the bound can either be zero or not. This leaves the question: does the size (of the bound) matter?

The answer is no. This can be seen easily if one considers the internal states as performing a gradient descent on surprisal. If the corresponding variational free energy has (approximately) the same value—to within an additive constant *c*—the dynamics will be identical everywhere (because the gradients do not depend upon the constant). In other words, it does not matter whether *c* is small or large: the approximate Bayesian inference interpretation only requires that the bound is (approximately) the same everywhere on the statistical manifold. See Figure 2.

This leads to the interesting question: are there any guarantees that the bound is constant? The answer is no. If it were possible to compute the bound, then one would use the surprisal or log evidence directly. In other words, the whole point of variational free energy is that it converts an intractable marginalization problem into a tractable optimization problem. Generally, one tries to optimize the form of the variational density to minimise variational free energy and thereby ensure a relatively tight bound that cannot vary substantially over the statistical manifold in question [19,42]. However, this is a purely practical consideration. From the point of view of self-evidencing, it just means that we can assert that self-organisation to nonequilibrium steady state necessarily entails exact Bayesian inference (i.e., self-evidencing) with posterior beliefs that are parameterized by something’s internal states. This inference may be exact; however, we will not be able to specify the form of posterior beliefs. Alternatively, the self-organisation can be interpreted as approximating approximate Bayesian inference under some parameterized form for the encoding of beliefs about the external states by internal states. With this distinction in place, we can now consider observation three.

## 5. Observation Three


*The gradients of the evidence bound vanish for nonequilibrium steady-state flows on the internal manifold.*


Biehl et al. [1] note that—for flow on the internal manifold—the gradients of the KL divergence or evidence bound disappear. This is true for both exact and approximate Bayesian inference interpretations. For exact Bayesian inference, the KL divergence is stipulatively zero everywhere, meaning the gradients vanish everywhere. For approximate Bayesian inference, the gradients of the KL divergence account for the difference between flow on the internal manifold and gradient flows on variational free energy. This difference accounts for the approximate equality in (10). Conceptually, this means that the free energy principle is not claiming that self-organisation to steady-state minimizes variational free energy; rather that self-organisation to steady-state can always be read as approximating approximate Bayesian inference. Practically, it means that if we specified a generative model (i.e., a desired steady-state density) and solved the following equations of motion (under an assumed form for the variational density) we can approximate self-organisation to a desired steady-state.
(12)$$\begin{array}{cc} f_{\alpha }\left(\mu ,b\right) & =\left(Q_{\alpha \alpha }-\Gamma_{\alpha \alpha }\right)\nabla_{\alpha }F\left(\mu ,b\right)\\ & \approx \left(Q_{\alpha \alpha }-\Gamma_{\alpha \alpha }\right)\nabla_{\alpha }ℑ\left(\mu ,b\right) \end{array}$$

Figure 3 illustrates a synthetic system, described in detail in [3] from which approximately inferential dynamics can be identified on finding the Markov blanket. In brief, the system in question comprises a set of macromolecules whose dynamics are governed by their electrochemical state and by Newtonian forces resulting from these states. On identifying a Markovian partition, the parametrisation of a variational density was based upon the maximally correlated linear combinations of the states of the internal and external macromolecules. This synchronisation gives the appearance that internal states are inferring external states: i.e., that the variational density—parameterised by internal states—is in the neighbourhood of the free energy minimum. The implication here is that inference may be thought of as (generalised) synchronisation across a Markov blanket.

## 6. Conclusions

In conclusion, we looked at three fundamental issues that underpin the free energy principle. The first was the role of solenoidal coupling—within and across the Markov blanket that defines anything of interest. The key observation—here and in Biehl et al. (ibid)—is that certain solenoidal coupling terms are precluded. One obvious example is the coupling between internal and external states. However, this does not necessarily preclude coupling between internal states and active states—or between external states and sensory states. Furthermore, there can be pronounced solenoidal coupling within any subset of the Markov blanket partition. The role of solenoidal coupling may be quite important in many systems. This is purely based on the heuristic that oscillatory and synchronous behavior underpins most biorhythms over many temporal scales and may be characteristic of biotic self-organisation [43,44,45]. In this setting, oscillations are assumed to be a manifestation of solenoidal flow; namely, circulation on iso-probability contours that form the fabric of classical (Lagrangian) mechanics (e.g., the orbits of heavenly bodies).

The second issue we looked at is the distinction between flows that look “as if” they are performing exact Bayesian inference, exactly or approximate Bayesian inference, approximately. The only thing that matters—in terms of this distinction—is if we want to parameterize and evaluate (or indeed simulate) posterior beliefs about external states that are parameterized by internal states. Crucially, these are not the internal states at any given moment. The internal states that constitute the statistical manifold are conditional expectations, given blanket states. This means that the interpretation in terms of Bayesian inference emerges only in expectation—or on average. This is an unremarkable and ubiquitous aspect of empirical studies of sentience. The classical example here is the averaging of multiple responses to sensory perturbations, when characterizing evoked responses in internal states (e.g., event-related potentials generated by internal neuronal states of the brain). See Figure 3. for an example. Since writing this paper, we have been encouraged by the enthusiasm with which these issues have been discussed in the literature. For readers interested in delving further into these exchanges, recent papers include [46,47,48,49]. In concluding, we would like to thank Biehl et al. for a thorough and useful deconstruction of [2].

## Figures and Tables

**Figure 1 entropy-23-01076-f001:**
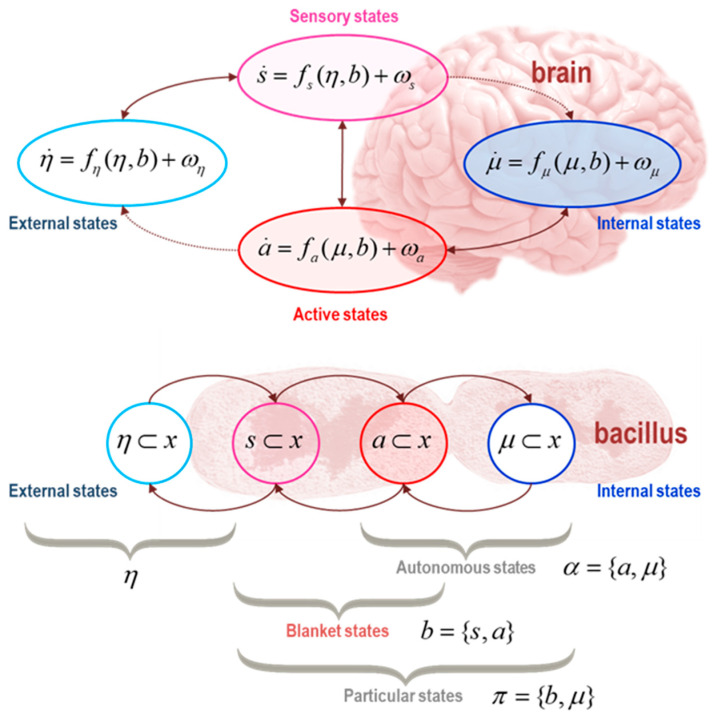
*Markov blankets*. This schematic (reproduced from [3]) illustrates the partition of states into *internal* states (*µ,* in blue) and hidden or *external* states (*η*, in *cyan*) that are separated by a Markov blanket (*b*) comprising *sensory* (*s*, in magenta) and *active* states (*a*, in red). The upper panel shows this partition as it would be applied to action and perception in a brain. Note that the only missing influences are between internal and external states—and directed influences from external (respectively internal) to active (respectively sensory) states. The surviving directed influences are highlighted with dotted connectors. In this setting, the self-organisation of internal states then corresponds to perception, while active states couple internal states back to external states. The lower panel shows the same partition but rearranged so that the internal states are associated with the intracellular states of a *Bacillus*, where the sensory states become the surface states or cell membrane overlying active states (e.g., the actin filaments of the cytoskeleton). Here, the coupling between sensory and internal—and between active and external states—was suppressed to reveal a simple coupling architecture that leads to a Markov blanket. *Autonomous* states (*α*) are those states that are not influenced by external states, while *particular* states (*π*) constitute a particle; namely, autonomous and sensory states—or blanket and internal states. The equations of motion in the upper panel underwrite the conditional independencies of the Markov blanket, as described in the main text. By this, we do not mean that these equations of motion define a Markov blanket, or even that they are necessary for a Markov blanket—which can exist in static systems with no dynamics. Instead, these equations represent flows that, under certain assumptions, result in a Markov blanket at steady state. Please see (1) and the associated discussion.

**Figure 2 entropy-23-01076-f002:**
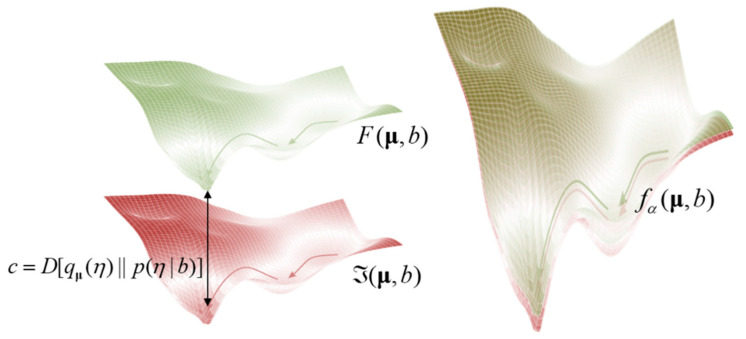
*Evidence bounds and gradient flows.* This schematic tries to convey the intuition that the gradient flows on surprisal (pink)—as a function of some statistical manifold (here conditional expectations of internal states, given blanket states)—are the same as gradient flows on variational free energy (green); if, and only if, the KL divergence or evidence bound is conserved over the manifold. The left panel shows a view of the two functions from the side, while the right panel provides a view from the top.

**Figure 3 entropy-23-01076-f003:**
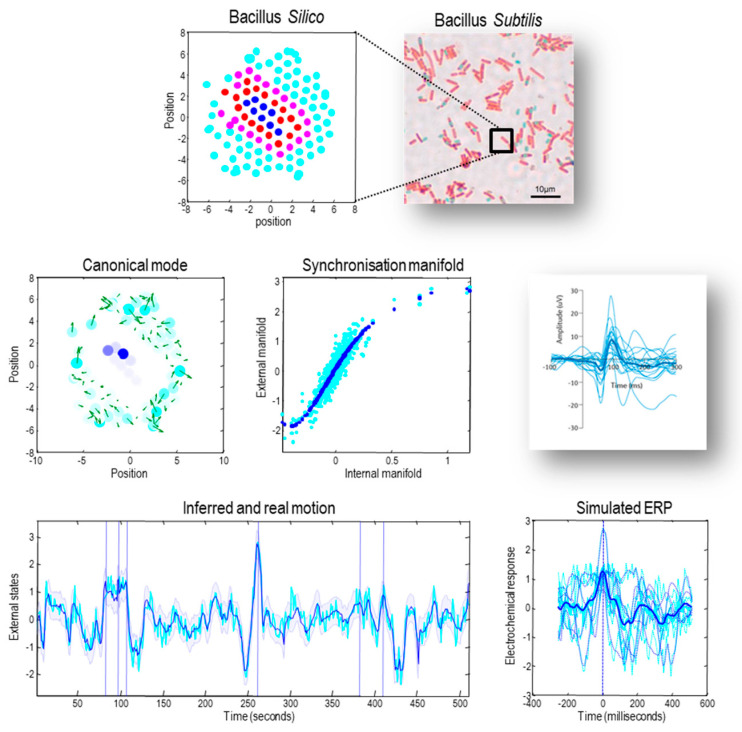
*Sentient dynamics and the representation of order*. This figure illustrates approximate Bayesian inference that follows when associating the internal states of a system with a variational (i.e., approximate posterior) density over external states. This figure is based upon the simulation of a small rod-like particle used to illustrate different perspectives on self-organisation in [3], where the details of this system are specified. In brief, each macromolecule is defined by a set of electrochemical states modelled as stochastic Lorenz attractor. These attractors are coupled between pairs of macromolecules, where the coupling strength depends upon the distance between each pair. In addition, the position and velocity of each macromolecule are described by (stochastic) Newtonian equations of motion subject to forces based upon the difference in electrochemical states between a molecule and its neighbours. The upper panels illustrate a collection of simulated macromolecules, in terms of internal (blue) active (red) sensory (magenta) and external (cyan) states. The middle left panel shows the first canonical vector of motion over the external states (green arrows) that are represented by the internal states (blue dots). The blue and cyan dots are placed at the location of internal and external states, respectively. The colour level reflects the norm (sum of squares) of the first canonical vectors showing the greatest covariation between external and internal states. The middle panel illustrates a synchronisation manifold (conditioned upon the Markov blanket) that maps from the electrochemical states of internal macromolecules to the velocity of external macromolecules. The blue dots identify the manifold per se, while the cyan dots are the estimated expectations used to estimate the manifold (using a fifth-order polynomial regression). The lower panel shows the same information but plotted as a function of time during the last 512 s of the simulation. The conditional expectation is based upon the internal states, while the real motion is shown as a cyan line. The blue shaded areas correspond to 90% confidence intervals. The lower right panel illustrates simulated event-related potentials of the sort illustrated by the insert (lower right panel). The simulated evoked response potential (ERP) was obtained by time locking the internal electrochemical states to the six time points that showed the greatest expression of the first canonical variate (indicated by the vertical lines in the middle panel). The dotted lines are six trajectories around these points in time, while the solid lines correspond to the average. The blue lines are the responses of internal states, while the cyan lines correspond to the real motion associated with the first canonical vector. The timing in the lower panels was arbitrarily rescaled to match empirical peristimulus times—illustrated with an empirical example of event-related potentials in the middle right panel.

## Data Availability

The software producing Figure 3 is available as part of the academic software SPM (https://www.fil.ion.ucl.ac.uk/spm/). It can be accessed by typing DEM and selecting the sentient physics button from a graphical user interface (**FEP_fluctuations.m**).

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
