# Peer review of "Some Interesting Observations on the Free Energy Principle"

_entropy, 2021, doi:10.3390/e23081076_

Round 1
Reviewer 1 Report
The present manuscript by Friston et al briefly reviewed the free energy principe and addressed a few technical issues raised by Biehl et al [2], including the critical role of solenoidal coupling in the free energy principle. I think the issues are fully addressed and the manuscript is well written. I suggest to publish as it is.
Reviewer 2 Report
A highly welcome clarification
Reviewer 3 Report
The authors discuss a recent article by Biehl, Pollock, and Kanai on the Free Energy Principle (FEP), and analyze some central points raised in the paper. In particular, they present three main observations related to solenoidal coupling terms, Kullback-Leibler divergence and gradients of evidence bound, relevant to the definition of the FEP.
In my opinion, the paper should be presented as a ``comment article'', rather than as an independent work. Indeed, without reading the cited article by Biehl et al, the reader cannot follow easily the discussion. In particular, the presentation of the FEP in section 2 is obscure, at least for a non-expert reader. First, many sentences, such as "anything that exists is actively seeking evidence for its existence" or "In short, life is its own existence proof", have more a philosophical taste, rather than scientific rigour. Second, the whole discussion including conditional independencies, Markov blanket
etc., does not help to clarify the meaning of the FEP, in the absence of some explicit examples. Third, the connection with integral fluctuation theorems is just mentioned without explanation and remains unclear. Therefore, in my opinion, in order to make clear the relevance and applications of the FEP, at least some explicit examples of physical systems should be provided and discussed in detail. Figure 1 somehow helps in the understanding of the general framework, but many symbols in the figure are not defined.
In section 4 there appears a claim that is not clear to me: indeed, Observation two states that "The difference between the variational
density and conditional density...", but at line 229 it is written "a
conditional (a.k.a., variational) density". So, if conditional and
variational density are synonyms, how there can be a difference
between the two? Please clarify this point. At line 214, KL clearly
stands for Kullback-Leibler, but should be explicitly said.
The quantities represented in Figure 3 are not clear to me. Could you please provide an explicit definition of the considered model? How do the simulated macromolecules interact? What defines the different states in the model? Some equation could help to understand.
Throughout the paper, many symbols remain undefined:
at line 113 is p(x) the stationary density of the system? Please define p(x);
at line 114, is I the identity matrix? Please define I;
at line 152, what is pi in the argument of f?
At line 535, what does ERP stand for?
At line 384, I think Martin should be cited by surname.
Overall, it is not clear what is new and what is reviewed in the paper. The authors should clarify which are the novel results reported in the article with respect to previous literature.
In conclusion, in my opinion, the paper is not suitable for publication in the present form.
Typos:
footnote 3: known known
Reviewer 4 Report
This manuscript is motivated by the technical critique made by Biehl et al (“A technical critique of some parts of the free energy principle”, subsequently referred to as TCFEP) to the influential paper “Life as we know it” (subsequently referred to as LAWKI). Personally, I believe this and Biehl’s manuscript are exemplar of a scholarly discussion with very positive scientific outcomes. In this regard, I congratulate the authors for engaging in this dialogue.
That being said, I believe the current manuscript can still be substantially improved. In the following, I’ll make a number of criticisms and suggestions, which are intended solely to encourage the authors to make the most of this response. I apologise in advance if some of my comments sound too harsh; as a matter of fact, I am a big fan of the FEP, and these comments just intend to further foster the technical development of the FEP.
Main concerns
1. TCFEP make a number of substantial critiques to the technical arguments developed in LAWKI; however, this response apparently addresses only a few of those. I believe it would be important to clarify why some of the points made in TCFEP are left unanswered.
2. One of the main points issued by TCFEP is the lack of clarity of the assumptions that are necessary and/or sufficient to make the results of LAWKI to hold. Unfortunately, this important issue is only partially touched in this response. In particular, the authors provide sufficient conditions for the Markov blanket condition to hold as a consequence of the specific restrictions on the flow, but necessary conditions are not discussed. More importantly, unless I my reading is mistaken, neither necessary nor sufficient conditions for the validity of the Free Energy Lemma are discussed. It is crucial to clarify these issues, in order for the readers to have a firm understanding of the domain of applicability of these results. Otherwise, all the comments about applicability of this formalism (to eg brains or bacilus) are unwarranted.
3. Another important point raised by TCFEP is the lack of clarity of how one should interpret the variational density q (discussed in detail in Sec 4 of TCFEP). Personally, I find this issue the biggest barrier to really understand what is going on in LAWKI, because in standard applications of the FEP one can interpret (as far as I know, I hope I am not mistaken) q as part of a process of internal modelling done by an agent, related to its counterfactual/imaginative capabilities. However, LAWKI puts forward an exiting attempt to interpret q in the context of eg dynamics purely physical processes. This is (in my opinion) a crucial move, but one that is not made as explicit as it could, considering its central role in the proposal. In particular, my understanding is that this works because there are a number of assumptions at play, but unfortunately those assumptions are not made clear in LAWKI, motivating the corresponding discussion found in TCFEP. This important issue is, unfortunately, not touched in this manuscript. In my opinion, it would be very helpful to clarify two things: (i) how the variational density should be understood in purely physical systems, and (ii) under what assumptions such interpretation give results of interest.
4. The level of technical detail employed in this manuscript is, in my opinion, at some points not entirely appropriate. In particular, given that this manuscript is directed to a readership that is interested and relatively proficient in the technicalities of the FEP, there are a number of issues that seem inappropriate:
a) Section 3 presents a high level recapitulation of the FEP without any technical details, but I am not sure what is the role that this plays in the current manuscript, given that this type of description can be found in other papers. I’d suggest to replace it by a more technical and formal recapitulation instead, or remove entirely and add a reference to other papers. Same with the explanation of the free energy presented after eq (1.11), including references to concepts (eg Lyapunov functions) that are not used later on. (As a side remark, considering that some variables in the KLs are averaged over and some don’t I’d suggest to simply write the Free energy explicitly — unless some of those decomposition are necessary for further derivations, which I fail to see).
b) There are numerous elements of notation that are introduced without being defined. For example: the function f in eq. (1.1), p in eq. (1.3), the subcomponents of f in eq. (1.9), and others. One can decipher all this by reading other FEP papers, but this way of procedure is not appropriate. Please ensure that each term in the equations has been explicitly explained in the text.
c) Some steps in the derivations are not made explicit, and is hard to follow. For example, from the explanation given in the text it is very hard to understand how the lower equation in (1.1) relates to the upper equation.
5. The clarification of the fact that the conditions imposed in the flow do not suffice to impose a Markov blanket structure in the stationary joint distribution is an important technical result achieved by the fruitful dialogue with TCFEP. In my opinion, papers focused on Markov blankets that actually don’t have a consistent technical definition of what exactly is a Markov blanket in the context of the FEP (eg “Markov blankets of life”, “Markov blankets, information geometry and stochastic thermodynamics”, and many others) do a disservice to the FEP in the eyes of mathematically oriented readers. This could be greatly improved if (i) this important omission in previous literature is explicitly acknowledged, (ii) a precise definition of what a Markov blanket means within the FEP is provided, and (iii) subsequent partial improvements (such as sufficient but not necessity conditions) are clearly accounted as such.
Additional comments / suggestions
1. Please note that (to the best of my knowledge) the Entropy journal does not allow footnotes. If this is the case, please provide a revised manuscript including the footnotes within the main text.
2. It is sometimes not trivial to track where in TCFEP are the specific points that are being replied here. In particular, the present manuscript refers to three “observations,” but they seem not to correspond to the “observations” enumerated in TCFEP. For example, I was not able to track where exactly in TCFEP is “observation two” of the current manuscript located (the potential unboundedness of the KL).
3. The opening text “all observations in TCFEP are interesting, some are correct and…” implies that some observations in TCFEP are technically incorrect. If you would like to keep this statement, please clarify which of the observations are mistaken, otherwise please re-write that text.
4. In pg., line 43: “a Langevin formulation of any random dynamical system”. To the best of my knowledge not any dynamical system can be described by Langevin dynamics. If this is correct, please consider re-writing this statement.
5. The text in lines 49-51 of pg. 2 suggests that no additional assumptions are needed in order to have a Markov blanket condition on the stationary distribution, which is one of the main points of later sections. Please consider clarifying this point.
6. Figure 1 perpetuates the conflation between the coupled stochastic differential equations and a Markov blanket that exists on its stationary distributions when additional conditions are put into place. Please be consistent with the points developed in the text and avoid referring to those equations as a “Markov blanket” to prevent future misunderstandings.
7. In line 107 pg 3, a decomposition of x into four components is presented, but it is not explained that those are supposed to denote internal, external, sensory and action variables. Please clarify this to aid the reader.
8. In line 139 pg 4, “In dynamical systems, the joint density is over states at different times”. I don’t understand what this statement refers to, please consider revisiting this passage.
9. In the derivation presented in Eq. (1.4), please highlight that, as noted in TCFEP, this only holds if the matrices Q and Gamma are constant.
10. In footnote 6, pg 5, it says that “the flows can be fine-tuned to crate Markov blankets in the absence of any sparsity constraints on coupling”. This is not trivial at least for me, please clarify how this could be done.
11. In line 181 pg 6, it says “there are four special cases of (1.7) that obtain when suppressing solenoidal coupling.” I don’t understand what does mean, please clarify.
12. In footnote 8 pg 6, the statement “the conditional independencies in (1.7) lead to the most general or canonical flow constraints”. Could you please clarify the meaning of this? These are the most general with respect to what? As it stands, this statement is (in my opinion) ambiguous.
13. In footnote 8 pg 6, it says that “these are simply counterexamples to the functional form of (1.7), not counterexamples that violate the assumptions of the free energy lemma”. Could you please explicitly refer to which counterexample is this comment is referring too? I failed to find such counterexample with explicit statements related to violations of the assumptions of the free energy lemma, but this may be my mistake.
14. The conditions described in eq. (1.9) depend on the form of the flow f, as it defines the form of Q and Gamma, is that correct? If so, could you please explain what types of flows give raise to (1.9)?
15. In line 218 pg 7, “In itself, this observation is unremarkable”. Can you please clarify what it is meant by this? Any observation in itself, without a context, is indeed unremarkable, no?
16. Please clarify under what conditions the approximation used in eq. (1.10) holds.
17. It seems that the footnotes have some problems with the references, see eg footnote 11 in pg 7.
18. In line 259 of pg 8, there is a mention of a conditional expectation, but I fail to see any. Could you please make this explicit?
19. I don’t see how Figure 3 is contributing to the text. It actually is not referenced. Please make its connection with the discourse more explicit, or remove.
Round 2
Reviewer 3 Report
The authors have addressed some of my previous comments, but not in a satisfactory way. In particular, I suggested to discuss explicitly an example to illustrate the general formalism, but the authors simply referred to another reference for applications to physical systems. My suggestion was motivated by the fact that, considering a specific simple system, also the technical discussions on the several points related to the paper by Biehl et al. would have been more clearly illustrated. As it stands, in my opinion, the discussion remains on a too general ground.
Concerning other minor points:
1) I do not see how sentences like "anything that exists is actively seeking evidence for its existence" or "life is its own existence proof" can help "to interpret the more technical material for a broader audience", as the authors write in their answer. In my opinion, these sentences have no meaning in a scientific context.
2) The clarification about the link with the Fluctuation Theorems is not accurate. I really would like to know how "arguments related to integral fluctuation theorems" could be used "to derive probability densities over the trajectory of active states", as the authors claim at lines 69-71. Moreover, in the same paragraph, the authors write "This allows one to characterize autonomous behavior (i.e., the dynamics of autonomous states) in terms of constructs from psychology and economics". Sincerely, I never found any application or relation of Fluctuation Theorems to psychology.
3) The difference between variational density and conditional density could be better illustrated with the help of an explicit example.
Overall, in my opinion, the paper is not suitable for publication in Entropy.
Reviewer 4 Report
The authors did a solid work on answering my observations and improving the manuscript. I believe the manuscript is almost ready for publication. Let me provide additional feedback about a couple of issues that I think could still be further improved.
1. I am not sure if the proposed definition of Markov blanket covers what the authors imply with the term. Right now, the definition of Markov blanket used is strictly aligned with Pearl’s, referring only to a statistical property of the variables in their steady-state. However, this condition is symmetric, and cannot account for the intended asymmetry in the relationships (eg that sensory states affect internal states but not vice-versa). It seems to me that this asymmetry is guaranteed by the specific underlying structure of the flow (having some zeros in the Jacobian), but this condition is neither necessary no sufficient for the Markov blanket condition. A valuable contribution of this manuscript is to provide sufficient conditions; however it would be important to provide a clarification about the relation (or lack of it) between the symmetry of the definition of the Markov blanket, and the asymmetry that is attributed to Markov blankets by many papers.
2. To one of my observations, the authors reply saying that “our aim is not to rehearse the arguments of LAWKI, which are now 8 years old, but to address those observations relevant to the field as it stands today.” I think this is an important point that serves to clarify the aim of the paper. However, I still have the concern that, given that the paper of Biehl et al raises numerous points and the authors here only comment on a few, some readers could get a feeling that the chosen issues have been cherry-picked in some way. Clarifying that the addressed points are the more relevant to the field as it stands today, while adding some more information of why the other points are no longer relevant, would greatly improve the manuscript. In particular, I wonder if some of these points that are not anymore relevant are related to other recent papers discussing related issues, for example [1-4]. Any clarifications that the authors could do in this lines I believe may help future readers to better understand this manuscript.
[1] Bruineberg, J., Dolega, K., Dewhurst, J., & Baltieri, M. (2020). The Emperor’s new Markov blankets.
[2] Raja, V., Valluri, D., Baggs, E., Chemero, A., & Aderson, M. L. (2021). The markov blanket trick: On the scope of the free energy principle and active inference.
[3] Aguilera, M., Millidge, B., Tschantz, A., & Buckley, C. L. (2021). How particular is the physics of the Free Energy Principle?. arXiv preprint arXiv:2105.11203.
[4] Da Costa, L., Friston, K., Heins, C., & Pavliotis, G. A. (2021). Bayesian mechanics for stationary processes. arXiv preprint arXiv:2106.13830.
2. In equation (1.1), from the notation it is not clear for me that the second line is a definition while the first is not (I actually thought this was a system of two equations). I suggest to re-write this in order to avoid similar confusions to other readers.
3. About the sentence “When this solution is expressed in terms of a Markov blanket partition…” The wording suggests (at least to me) that one can always partitionate the steady state into a Markov blanket form. I believe it could be helpful to clarify that this is not always possible, eg “In case the solution can be expressed in terms of a Markov blanket partition…”
4. A final, side question related to the new text that refers to Figure 3. Is inference equivalence to synchronisation, and is the framework proposing a re-interpretation of synchronisation? Or is inference a subset of synchronisation phenomena? It could be interesting to clarify the relationship between these two broad classes of phenomena (most likely not in this paper...)
Round 3
Reviewer 3 Report
I really thank the authors for their efforts to address the raised points. However, I am not satisfied by their answers. I am a theoretical physicist expert in the field of non-equilibrium statistical mechanics and, sincerely, the whole discussion remains obscure to me. I suggested to present a simple example to illustrate the several steps underlying the general theory. The authors mentioned a thermometer, without providing any explicit equations. So the discussion remains too vague. I would have appreciated the analysis of a specific (simple) system described by a mathematical model, with the identification of the several quantities appearing in the discussion. Also, the predictive power of the general approach is not clear without some explicit and specific examples.
Therefore, from my perspective, the article is not suitable for publication.